# The Identifiability of Prefix Control in Large Language Models

## Abstract

Current practice in prompting, evaluation, and alignment of large language models (LLMs) often treats behavioural similarity as evidence of similar underlying control. This assumption is rarely examined at the level where control is instantiated at the human–model interface. Across diverse decoder-only models, fixing the prefix yields separable final-layer representations immediately prior to decoding, even when the continuation varies. Using a simple centroid-based criterion on final-layer interface vectors, prefix identity can be recovered with high accuracy in our evaluated setup, with results varying across models.

## 1 Introduction

Most interaction with large language models takes place through short pieces of text. Prefixes are typically interpreted as instructions whose meaning is expected to guide the model's response, and output similarity is often treated as evidence of similar control.

Such inferences are based almost entirely on observable behaviour, without examining how control is realised at the interface between input and generation. This is especially common in offline or on-device settings and in domain-restricted deployments, where interactive correction is limited Xu et al. (2024); Amodei et al. (2016); Voigt & von dem Bussche (2017). In these settings, behaviour is often used as a proxy for control stability even though the representation state at generation time remains unobserved.

Across different models, short textual prefixes induce distinct regions in final-layer representation space immediately prior to decoding, where representations are most directly tied to token prediction Vaswani et al. (2017); Rogers et al. (2020). When the prefix is held fixed, these interface representations cluster reliably despite substantial variation in the continuation.

**A missing assumption in the instruction-following view.** This way of describing prompting assumes that equivalent intent corresponds to similar internal states. Such an assumption reflects a long-standing symbolic view of instruction and action, where instructions are treated as plans to be interpreted and executed Agre (1997); Suchman (1987). From a situated action perspective, behaviour that appears to follow instructions does not necessarily rely on the execution of an explicit plan Brooks (1991); Clark (1997); Glenberg (1997).

A prefix does not operate by specifying an instruction to be executed. It biases the conditions under which generation begins. In Shannon's formulation, a signal does not encode intent, it constrains the space of possible responses Shannon (1948). At the interface level, analysis is necessarily limited to observable behaviour, without committing to any particular account of the underlying mechanisms Marr (1982); Norman (1986).

**Why geometry, and why "regions"?** Neural computation has long been understood in terms of distributed representations, where meaning is expressed as patterns in a high-dimensional state space Rumelhart & McClelland (1987). Subsequent work in neurocomputational modelling and systems neuroscience has taken this view further. Neural activity is not arbitrary. Instead, neural activity tends to organise into stable pat-

terns in high-dimensional space, such as regions, trajectories, and attractor-like regimes Churchland (1989); Amit (1992); Sussillo & Barak (2013).

Modern representation learning follows a similar geometric intuition. In representation learning, features are not arbitrary but tend to form structured patterns in high-dimensional spaces Bengio et al. (2013). Geometric deep learning follows this line of work by viewing representations as structured objects in high-dimensional space, for which manifold-based descriptions are often useful Bronstein et al. (2017); Cunningham & Yu (2014); Vyas et al. (2020). Here, we use the term region to describe a recurring and separable organisation in final-layer representation space induced by a fixed prefix under our protocol.

**Terminology.** Throughout, we use the term "region" in a purely descriptive sense, to refer to a recurring and separable subset of final-layer representation space associated with a fixed prefix under our extraction and scoring protocol. We do not assume low dimensionality, smoothness, connectivity, or any manifold structure, and none of our results rely on such assumptions.

**Interface-level control state.** We define the *interface-level control state* as the final-layer representation vector $z_{\pi,x}$ immediately prior to decoding. This vector fully determines the probability distribution over the next token and thus governs all subsequent generation. Two inputs are said to share the same control state if their $z$ vectors are assigned to the same prefix centroid under our nearest-centroid criterion.

For this reason, our analysis stays at the human–model interface. Users interact with the model only through inputs and outputs, while internal processes remain unobservable and vary across architectures. We focus on the final layer immediately before decoding, where representations are most directly tied to generation and remain comparable across architectures Elhage et al. (2021); Skean et al. (2025). This layer forms the shared output interface of autoregressive models and directly determines token predictions.

This choice follows from a simple property of autoregressive generation. Although Transformer architectures are structurally symmetric, generation is inherently asymmetric, as generated tokens condition later computation. Prefixes therefore have a lasting effect on generation. If prefixes determine response regimes, their impact should be long-lived and visible at the output level, rather than as short-lived internal activations.

We examine this effect by fixing the prefix and varying the subsequent continuations, focusing on settings that preserve semantic intent, such as summarisation and compression. Our analysis is based on representations extracted at the final layer. With the prefix held fixed, representations consistently occupy a separable region in our evaluated setup across models and languages. The resulting structure is separable under our evaluation protocol; even simple geometric comparisons, such as cosine similarity to prefix centroids, are sufficient to identify the governing prefix.

Similar outputs do not imply similar representations Tennenholtz et al. (2024); Nanda et al. (2023). A geometric perspective therefore provides a more natural way to describe prompt behaviour. Properties such as robustness and cross-model transfer are therefore more naturally examined at the level of representation structure, rather than through output-level evaluation alone.

## 2 Methods

### 2.1 Interface level perspective

We study prefix based control as a *communication* phenomenon defined at the human–model interface. Users interact with models through inputs and judge their behaviour from the outputs, while everything inside the model remains opaque. We therefore focus on how short prefixes affect representations near the point of generation, in a way that can be compared across model families.

For each model $M$, we use the same setup. This setup defines a fixed set of short prefixes $\mathcal{P}$ paired with a shared set of texts $\mathcal{T}$ under a unified formatting rule. Prefix-conditioned representations are then extracted for each $(\pi, x)$ and analysed to determine whether they cluster by prefix using a non-parametric geometric criterion. No fine-tuning, prompt tuning, or task supervision is used at any stage.

## 2.2 Language models

We evaluate publicly released decoder-only Transformer language models spanning multiple families and scales, including GPT-2, LLaMA-family, Qwen-family, Phi-family, and Mistral-family models. Unless stated otherwise, we use pretrained base checkpoints. All models Radford et al. (2019); Grattafiori et al. (2024); Yang et al. (2025); Abdin et al. (2024); Jiang et al. (2023); Mistral AI (2025) are run in inference mode with hidden-state outputs enabled.

## 2.3 Prefix signals

We use a small set of short textual prefixes $\mathcal{P}$, including both declarative statements and instructional prompts. These prefixes include multilingual variants (e.g., English, Chinese, Japanese, Korean, Arabic, Hindi, French, German, and Spanish), as well as intentionally noisy or symbolic variants. Prefixes are short—typically fewer than ten tokens—and are not tuned to any specific model. We treat each prefix as its own condition. By "functional equivalence," we mean equivalence at the level of human-judged output behaviour. At the model level, prefixes are not instructions but different token sequences, distinguished by their wording and placement.

In total, we use $|\mathcal{P}| = 25$ fixed prefixes, including English strings, Traditional and Simplified Chinese variants, multilingual paraphrases, and controlled noise injections (including symbol and emoji variants). The prefix set is held constant across all models and all input texts. The complete prefix list, together with language and functional labels used in the main figures, is provided in Appendix C.

**Prompt injection test.** We run a separate prompt injection test that is independent of the prefix-set experiments (i.e., it introduces an additional injected string but uses the same representation and scoring pipeline). **Task:** insert a short placeholder string (we use `financial`) into the input at a fixed position. **Competing:** the same input without the injected string. Concretely, we construct the base input as $\pi \,\|\, \text{"\textbackslash n"} \,\|\, x$ and then insert `"\nfinancial\n"` immediately after the prefix line and before the content segment $x$, keeping this rule fixed across all samples and models. We compare the two conditions under the same nearest centroid rule and report the centroid margin (top1 minus top2 cosine) under each condition.

## 2.4 Shared input texts

We use the same set of natural-language texts $\mathcal{T}$ for all prefixes within each model. Inputs are truncated to a fixed maximum length $L$. $\mathcal{T}$ is drawn from the public `Salesforce/wikitext` dataset [1], using the `wikitext-103-raw-v1` split. We use the raw text as provided by the dataset and do not apply additional cleaning, normalization, or filtering beyond standard tokenization and length truncation. Short and long regimes are defined by the character length of the input content $x$: $\text{length(x)} < 100$ (short) and $\text{length(x)} > 1000$ (long).

**Concrete input formatting.** Each evaluation input is constructed as a raw text string by concatenating a prefix and a content segment: $\text{STR} = \pi \,\|\, \text{"\textbackslash n"} \,\|\, x$, where $\pi \in \mathcal{P}$ and $x \in \mathcal{T}$. Independently of the prompt injection probe, we optionally append a fixed probe suffix `"\n[PROBE]"` to the end of the raw string: $\text{STR} = \pi \,\|\, \text{"\textbackslash n"} \,\|\, x \,\|\, \text{"\textbackslash n[PROBE]"}$. The resulting string is then tokenized using the model specific tokenizer; special tokens such as BOS are handled by the tokenizer and model configuration.

All constructed strings `STR` are tokenized using the model-specific tokenizer and truncated to a maximum length $L = 512$ tokens. If truncation occurs, the effective content length and truncation flag are recorded in an experiment manifest for reproducibility.

## 2.5 Representations at the interface

**interface-level focus.** We use representations from the final Transformer block, prior to decoding. This layer provides a shared point of comparison across autoregressive models, whereas earlier layers differ sub-

---

[1] https://huggingface.co/datasets/Salesforce/wikitext

stantially across architectures. Our goal is not to explain how internal representations are formed, but to examine how prefix-based signals appear at the interface where inputs turn into outputs.

Let $Z_{\pi,x}^{(\ell)} \in \mathbb{R}^{n \times d}$ denote the hidden states at layer $\ell$, with $n$ the sequence length of the processed input sequence (after optional probe suffix) and $d$ the hidden dimension. We use the output of the final Transformer block, $\ell = \ell_{\text{last}}$, as our representation. (corresponding to `hidden_states[-1]` in our implementation). Specifically, we extract a vector $z_{\pi,x} \in \mathbb{R}^d$ from a fixed token position at the interface (in all experiments reported in this paper, we use the last token of the processed input sequence; let $n_{\text{proc}}$ denote the processed sequence length) ,

$$z_{\pi,x} = Z_{\pi,x}^{(\ell_{\text{last}})}[n_{\text{proc}} - 1].$$

All representations are $\ell_2$-normalised prior to comparison. We $\ell_2$-normalise prefix centroids prior to cosine similarity.

## 2.6 Centroid construction

For each prefix $\pi$, we summarise its typical representation by averaging its vectors over a *centroid-construction split* of independent texts,

$$\bar{z}_\pi = \frac{1}{|\mathcal{T}_{\text{con}}|} \sum_{x \in \mathcal{T}_{\text{con}}} z_{\pi,x},$$

where $\mathcal{T}_{\text{con}} \subset \mathcal{T}$ is a randomly selected subset (90% of texts by default). All evaluation is performed on a disjoint held-out split $\mathcal{T}_{\text{eval}} = \mathcal{T} \setminus \mathcal{T}_{\text{con}}$.

**Clarification on what is averaged.** The centroid $\bar{z}_\pi$ averages *only across texts $x \in \mathcal{T}_{\text{con}}$*. Each term $z_{\pi,x}$ is a single vector extracted at the fixed interface token position from the final Transformer block (Section 2.5).

## 2.7 Geometric separation of prefixes

For each held-out representation $z_{\pi,x}$ with $x \in \mathcal{T}_{\text{eval}}$, we compare it against all prefix centroids using cosine similarity,

$$\text{sim}(\pi' \mid z_{\pi,x}) = \cos(z_{\pi,x}, \bar{z}_{\pi'}) = \frac{z_{\pi,x}^\top \bar{z}_{\pi'}}{\|z_{\pi,x}\|_2 \|\bar{z}_{\pi'}\|_2}.$$

Each representation is matched to the prefix with the highest centroid similarity,

$$\hat{\pi} = \arg\max_{\pi' \in \mathcal{P}} \text{sim}(\pi' \mid z_{\pi,x}).$$

We evaluate performance using three quantities: top1 prefix identification accuracy (`acc_top1`), the average similarity to the corresponding prefix centroid (`avg_cos_top1`), and the gap between the highest and second-highest centroid similarities (`avg_margin`).

## 2.8 Stability across data splits

To check that the results are not sensitive to a particular split, we repeat the train–test partitioning with multiple random seeds and report the mean and standard deviation. We also include a simple sanity check by randomly permuting prefix labels before centroid assignment. After permuting the labels, accuracy drops to near chance (approximately $1/|\mathcal{P}|$). This suggests that the separation does not come from the evaluation procedure itself.

**Implementation details.** All experiments are inference-only. For each $(\pi, x)$, we run a single frozen forward pass with hidden-state outputs enabled and extract the final-block interface vector from a fixed token position (last token by default). We log tokenization lengths, truncation flags, and the saved layer index for each sample to ensure reproducibility across models and tokenizers. Unless otherwise specified, model inference uses `float16` on GPU and `float32` on CPU, while centroid construction and cosine-similarity evaluation are performed in `float32` on CPU.

# 3 Results

Our results concern control at the interface rather than task performance. Figs. 1 and 2 show how prefix-induced structure appears across models, languages, and control settings under a fixed evaluation setup.

## 3.1 Fixed prefixes lead to separable final-layer representations

Final-layer representations retain sufficient structure for prefix identity to be recovered with high accuracy under short inputs and above-chance accuracy under long inputs.

## 3.2 Interface control under interference

We next examine whether the interface-level organisation remains well-resolved once an additional injected string is introduced. Here, *interference* refers to inserting a short placeholder string (we use `financial`) into the input text under the fixed rule described in Methods.

In this prompt injection probe, we compare inputs with and without the injected string under the same representation choice ($z_{\pi,x}$ from `hidden_states[-1]`) and the same nearest-centroid scoring rule. Separation is reduced under **Task** (with injection), reflected by a smaller top1-top2 centroid margin. The generated outputs can appear unchanged under coarse behavioural inspection even when the interface state becomes less clearly associated with a single centroid under the same rule.

To test whether this effect generalises beyond a single model or language, we examine the same joint condition across model families, parameter scales, and languages. Fig. 2 reports the corresponding accuracy patterns across models under the same evaluation protocol, while the margin statistics are summarized separately in Fig. 3.

This reveals a mismatch between behavioural stability and interface-level control stability. A model can continue to produce task consistent outputs even after leaving the original prefix regime. From the outside, nothing appears wrong, yet the representation state that governs generation has already shifted.

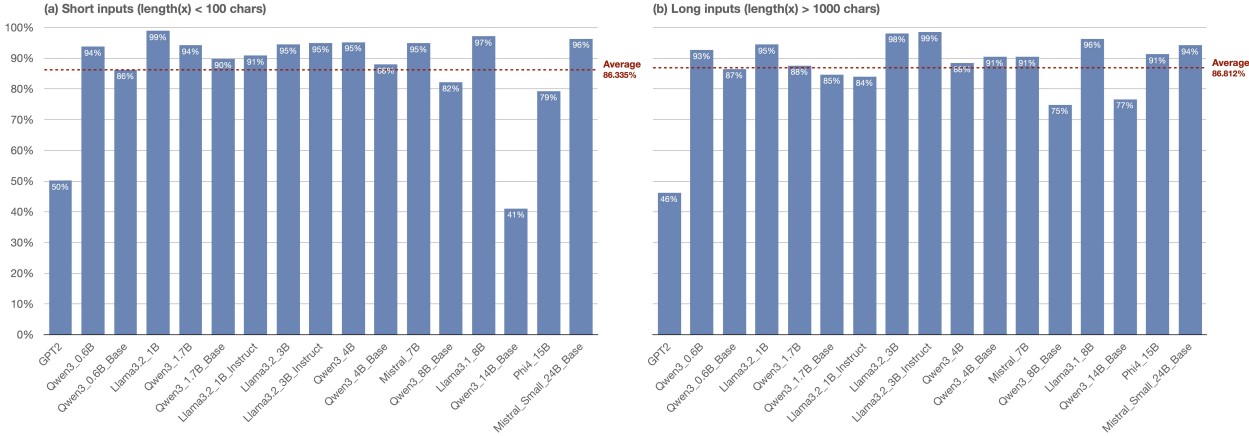

Figure 1: **Prefix identification accuracy under short and long input regimes.** Top1 accuracy of nearest-centroid prefix identification using cosine similarity between final-layer interface vectors and prefix centroids. Short and long regimes are defined by the character length of the input content $x$: `length(x) < 100` (short) and `length(x) > 1000` (long). The dashed line indicates the mean accuracy across models within each panel.

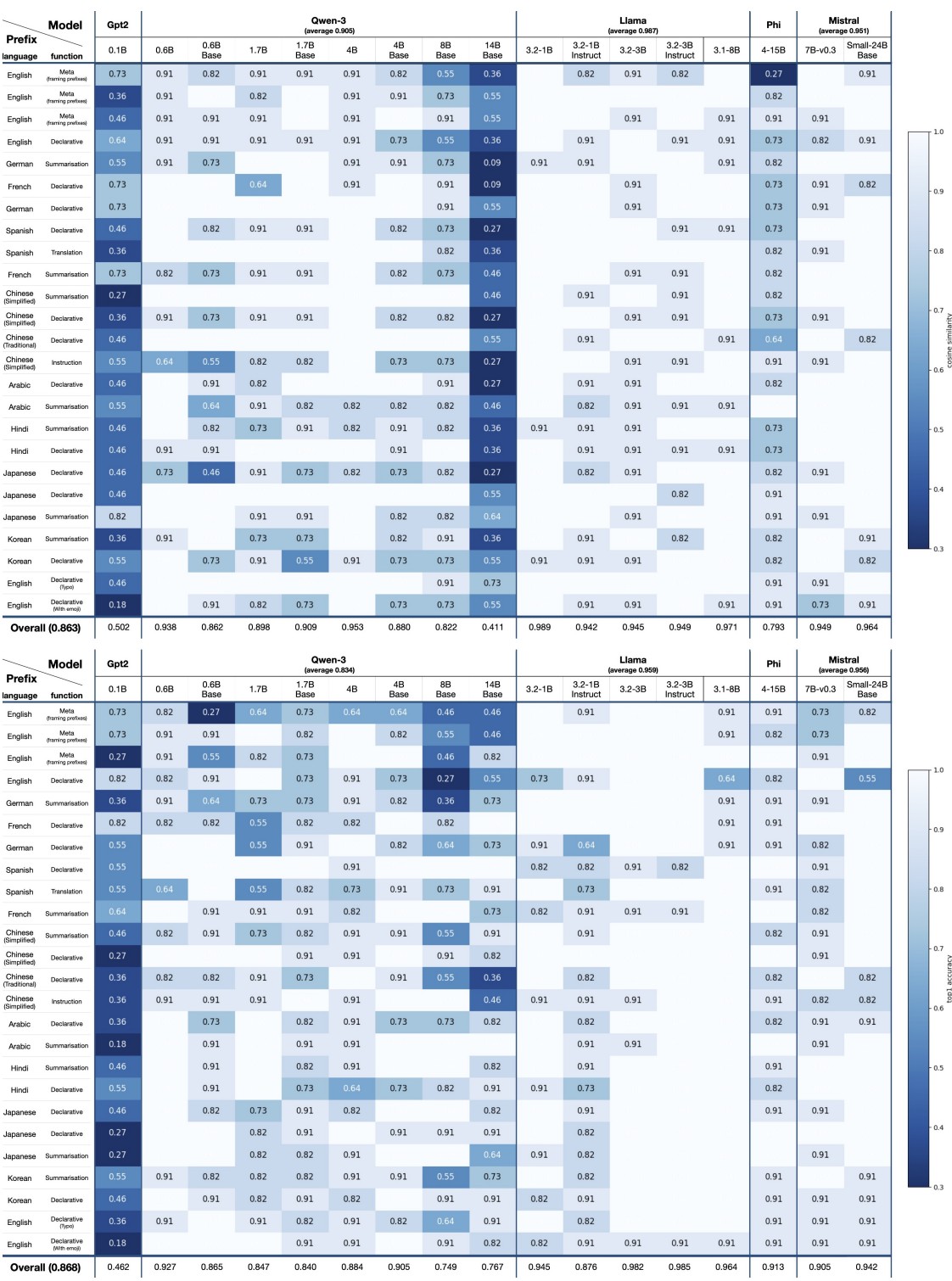

Figure 2: **Prefix-dependent structure at the model output interface.** Top1 prefix identification accuracy (nearest-centroid classification) computed from final-layer interface representations (`z_last`). **Top panel:** Short inputs (`length(x) < 100` characters). **Bottom panel:** Long inputs (`length(x) > 1000` characters). Blank cells indicate `acc_top1 = 1.00` and are left unannotated for readability. All shown model checkpoints and prefix conditions were evaluated under the same protocol.

### 3.3 Prompt injection reduces state resolution

We next examine whether these interface-level control states remain stable when a short injected string is inserted into the input text (we use `financial` as a placeholder). Rather than contrasting clean and attacked inputs, we track how the interface state changes under this joint condition. The interface state may remain well-resolved under the nearest-centroid rule, or it may become ambiguous once the injected string is present.

We observe that inserting the injected string systematically reduces geometric separation across models. The difference between the top two prefix matches becomes much smaller, leaving the interface state without a clear association. Although behaviour can appear consistent with the task under coarse output-based checks, final layer representations exhibit markedly reduced margins relative to task-conditioned centroids. In many cases, representations lie close to multiple prefix-induced regions, yielding low-margin (ambiguous) regime assignments.

From an evaluation perspective, these cases are consequential precisely because they can pass behavioural spot checks. With even minimal interference, the model can leave the original control state while its outputs still look unchanged. From the outside, nothing appears wrong, but the interface state has already shifted.

Fig. 3 summarizes the margin statistics across models for the two conditions. This effect is observed across model families. These differences are not visible from output behaviour alone, but emerge clearly at the interface level under the same scoring rule.

## 4   Discussion

**State resolution under the prompt injection probe.**   Fig. 3 and Table 1 summarise how the injected string affects state resolution at the interface under the same scoring rule. The decisive factor is the control state present at decoding, rather than whether task behaviour appears unchanged.

**Scope of claims.**   Our contribution is to clarify what constitutes valid evidence for inferring control in human and LLM interaction. No new prompting or training method is proposed, and no mechanism level claims are made.

While behavioural evaluation remains indispensable, our results motivate complementing it with interface-level measurements at decoding time. Accordingly, we evaluate control, robustness, and stability at the output interface, without assuming internal mechanisms.

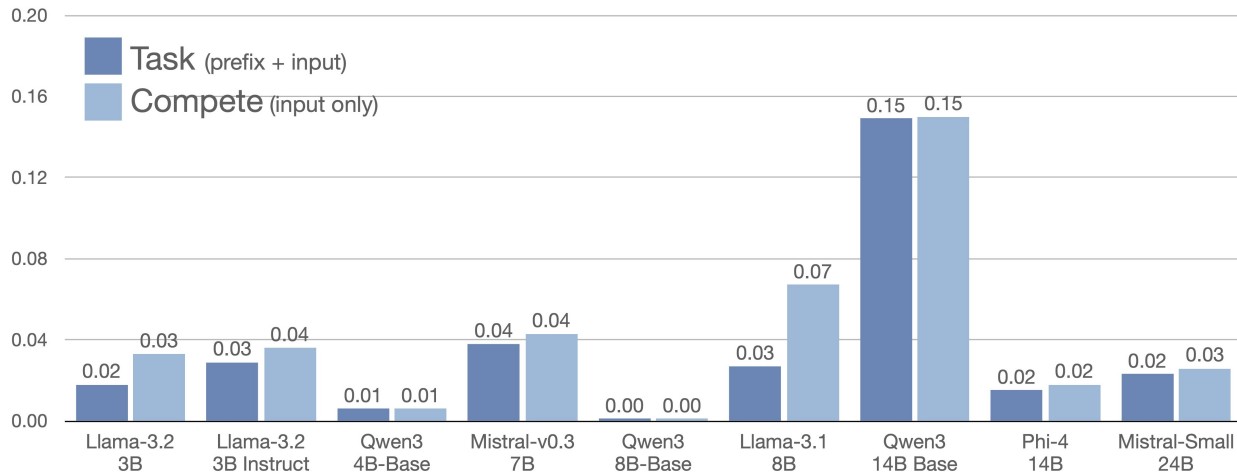

Figure 3: **Prompt injection effects at the interface.** Change in centroid margin across models when the injected string is present versus absent. **Task** inserts the injected string (we use `financial`) into the input text; **Competing** uses the same input text without the injected string.

Table 1: **Prefix resolution under the prompt injection probe. P**: approximate parameter count. **Acc**: top1 nearest-centroid prefix identification accuracy. **Mar**: centroid margin (top1 minus top2 cosine). $\Delta$Mar reports $\text{Mar}_{\text{comp}} - \text{Mar}_{\text{task}}$. **Task** inserts the injected string (we use `financial`) into the input text; **Comp** uses the same input text without the injected string.

| Model | P | $\text{Acc}_{\text{task}}$ | $\text{Acc}_{\text{comp}}$ | $\text{Mar}_{\text{task}}$ | $\text{Mar}_{\text{comp}}$ | $\Delta$Mar |
|---|---|---|---|---|---|---|
| Qwen3-4B-Base | 4B | 0.700 | 1.000 | 0.006 | 0.006 | 0.000 |
| Qwen3-8B-Base | 8B | 1.000 | 1.000 | 0.001 | 0.001 | 0.000 |
| Qwen3-14B-Base | 14B | 0.700 | 0.600 | 0.149 | 0.150 | 0.001 |
| Llama-3.1-8B | 8B | 0.800 | 0.900 | 0.027 | 0.067 | 0.040 |
| Llama-3.2-3B | 3B | 1.000 | 0.900 | 0.018 | 0.033 | 0.015 |
| Llama-3.2-3B-Instruct | 3B | 1.000 | 1.000 | 0.029 | 0.036 | 0.007 |
| Phi-4 | 14B | 1.000 | 1.000 | 0.015 | 0.018 | 0.003 |
| Mistral-7B-v0.3 | 7B | 1.000 | 0.900 | 0.038 | 0.043 | 0.005 |
| Mistral-Small-24B | 24B | 1.000 | 1.000 | 0.023 | 0.026 | 0.003 |

## 4.1 The effects of prefix across models

All models are evaluated under a shared interface. Prefix effects vary substantially across models, indicating that the observed structure cannot be explained by a shallow output-level mapping alone. Among recent models, separation is strongest at mid-range scales, while at larger scales the structure becomes less distinct. Higher capacity may allow greater interaction between conditioning signals.

Prefix identity remains above chance, suggesting continued but diminished prefix related structure. Reduced separation in representation space does not necessarily manifest as a change in outputs. Models can continue to behave normally and score well on benchmarks even after their control state has shifted. These patterns are observed under the same prefix setting across models, although the strength of the effect differs across architectures.

## 4.2 How prefixes shape representation

This kind of behaviour shows up routinely in use. A prefix does not trigger an instruction so much as it sets the state the model starts from when generating a response.

**Prompt robustness as regime stability.** Small edits in a prefix can move the model between nearby response states. When this happens, output behaviour may remain unchanged even though the control state has already drifted.

**Why cross-model prompt transfer can fail.** Two models can produce similar outputs for the same prompt, yet occupy different regions of representation space. A prompt can work reliably in one model yet fail in another, even when the outputs look the same.

**Conclusion** Prompting in large language models is commonly described as instruction following. Across model families, parameter scales, and languages, short prefixes induce identifiable and separable regions in final-layer representation space immediately prior to decoding. Prefix identity remains distinguishable in final-layer representations with high short-regime accuracy and above-chance long-regime accuracy using a simple geometric criterion, even under substantial variation in continuation. Yet, prefixes that produce comparable outputs can correspond to distinct regions of representation space.

**Implications for evaluation and inference.** Outputs that look the same can arise from different control states. When evaluation relies only on behaviour, this difference is easy to miss. Success judged from outputs alone does not guarantee that the model is being controlled in the same way. At the interface, prompting functions as a way of setting the model's starting state, not as an instruction to be carried out.

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

## A  Evaluated Model Checkpoints

| Checkpoint | Family | Approx. Params |
|---|---|---|
| openai-community/gpt2 | GPT-2 | 0.1B |
| meta-llama/Llama-3.2-1B | LLaMA 3.2 | 1B |
| meta-llama/Llama-3.2-1B-Instruct | LLaMA 3.2 | 1B |
| meta-llama/Llama-3.2-3B | LLaMA 3.2 | 3B |
| meta-llama/Llama-3.2-3B-Instruct | LLaMA 3.2 | 3B |
| meta-llama/Llama-3.1-8B | LLaMA 3.1 | 8B |
| Qwen/Qwen3-0.6B-Base | Qwen3 | 0.6B |
| Qwen/Qwen3-0.6B | Qwen3 | 0.6B |
| Qwen/Qwen3-1.7B-Base | Qwen3 | 1.7B |
| Qwen/Qwen3-1.7B | Qwen3 | 1.7B |
| Qwen/Qwen3-4B-Base | Qwen3 | 4B |
| Qwen/Qwen3-4B | Qwen3 | 4B |
| Qwen/Qwen3-8B-Base | Qwen3 | 8B |
| Qwen/Qwen3-14B-Base | Qwen3 | 14B |
| microsoft/phi-4 | Phi | 14B |
| mistralai/Mistral-7B-v0.3 | Mistral | 7B |
| mistralai/Mistral-Small-24B-Base-2501 | Mistral | 24B |

Table 2: All model checkpoints evaluated in this study.

## B  Concrete Input Example

For completeness, we provide one literal instantiation of the input construction rule described in Section 2.5.

**Example**

```
PREFIX:
Veuillez résumer le texte suivant

CONTENT (excerpt):
The history of machine learning began with early
attempts to formalize pattern recognition and
statistical inference. In the mid-twentieth century...

FULL INPUT (used in forward pass):
Veuillez résumer le texte suivant
The history of machine learning began with early
attempts to formalize pattern recognition and
statistical inference. In the mid-twentieth century...
[PROBE]
```

Formally, we construct a raw text string as

$$\texttt{STR} = \pi \,\|\, \text{newline} \,\|\, x,$$

and when the probe is enabled we append a fixed probe suffix at the end of the string

$$\texttt{STR} = \pi \,\|\, \text{newline} \,\|\, x \,\|\, \texttt{"\textbackslash n[PROBE]"}.$$

The final token sequence is obtained by applying the model specific tokenizer to STR; special tokens such as BOS are handled by the tokenizer and model configuration.

## C  Prefix Set

| prefix | languages | function |
|---|---|---|
| Machine Learning | English | Meta (framing prefixes) |
| How Prefixes Shape the Manifold | English | Meta (framing prefixes) |
| The Geometry of Communication Between Humans and LLMs | English | Meta (framing prefixes) |
| this is an experiment | English | Declarative |
| expirement | English | Declarative (Typo) |
| ✏️ experiment | English | Declarative (With emoji) |
| Ceci est une expérience | French | Declarative |
| Veuillez résumer le texte suivant | French | Summarisation |
| Dies ist ein Experiment | German | Declarative |
| Bitte fassen Sie den folgenden Text zusammen | German | Summarisation |
| Esto es un experimento | Spanish | Declarative |
| Traduce el siguiente texto al chino | Spanish | Translation |
| 这是一个实验 | Chinese (Simplified) | Declarative |
| 请总结以下内容 | Chinese (Simplified) | Summarisation |
| 這是實驗 | Chinese (Traditional) | Declarative |
| 請用正式學術語氣回答以下內容 | Chinese (Traditional) | Instruction |
| 【実験】 | Japanese | Meta (framing prefixes) |
| これは実験です | Japanese | Declarative |
| 以下の文章を要約してください | Japanese | Summarisation |
| 이것은 실험이다 | Korean | Declarative |
| 다음 내용을 요약하시오 | Korean | Summarisation |
| هذه تجربة | Arabic | Declarative |
| يرجى تلخيص النص التالي | Arabic | Summarisation |
| यह एक प्रयोग है | Hindi | Declarative |
| निम्नलिखित पाठ का सारांश दें | Hindi | Summarisation |

Figure 4: Complete prefix list with language and functional annotations.

