# OpenReview forum: "The Identifiability of Prefix Control in Large Language Models"
_TMLR — Rejected by TMLR_

### Review · Reviewer_pUEv · 2026-02-05

**Summary Of Contributions:**

The paper attempts to prove the claim that prefixes/"system prompts" partition the feature space such that outputs with the same prefix have similar representations even though they have potentially different content, and meanwhile outputs with different prefixes may have different representations even though they have potentially similar content, i.e., the representation is more aligned with the prefix than the suffix/output.  They show this by assessing a variety of 100M-4B parameter open source models on some prompts with a variety of system prompts, computing the centroid of the features amongst all prompts with the same system prompt, and computing statistics of the similarity of each feature against its centroid and nearby centroids (and the margin between each pair of centroids, etc). They also study what happens in feature space if competing instructions are provided to the system prompt (in which case the margin shrinks, etc).

**Audience:**

Yes

**Audience Explanation:**

Understanding the feature space of LLMs is an area of interest in the ML community, especially those working on interpretability and alignment. There is therefore some overlap of the paper with the interests of the community.

**Broader Impact Concerns:**

No broader impact concerns

**Claims And Evidence:**

No

**Claims Explanation:**

- The paper is extremely imprecise in its writing and presentation of scientific results. For example:
  - Many non-standard terms are frequently used without any attempt at definition ("interface", "control", etc), so it is often difficult to understand what the precise scientific claims are. On a more aesthetic note, it would be good to state the claims up-front.
  - No datasets are specified, only abstract "Tasks" (no examples of prompts, instructions). On the other hand, the models are specified.
  - Some figures need to be made more precise, e.g., Figure 4 y-axis seems to be going up in increments of 0.08 and not 0.1, meanwhile the y axis has two labels saying (rounding to) 0.2.
  - The notion of "similar output" seems to be human-defined and not precisely reproducible (see Section 2.2). This makes the conclusions more unreliable and suspicious, especially when there are not examples of the outputs or even specifications of the datasets and models. Similar for "competing directives" in the system prompt.
  - The paper does not actually specify how to collect the features. In particular, Section 4.5 says that either of the following mechanisms can be used: $z\_{\pi, x} = Z\_{\pi, x}^{\ell\_{\mathrm{last}}}[n]$ OR $z\_{\pi, x} = \frac{1}{n}\sum_{i = 1}^{n}Z\_{\pi, x}^{\ell_{\mathrm{last}}}[i]$, but what definition is used for which experiment is left unspecified.
  - Overall, the description of the hypothesis, each experiment, the description of the results, and the implications are sorely lacking of almost all detail required to understand whether they support the overall claim (and also whether the experimental design is scientifically valid/reproducible).

- Supposing that the averaging mechanism $z\_{\pi, x} = \frac{1}{n}\sum_{i = 1}^{n}Z\_{\pi, x}^{\ell_{\mathrm{last}}}[i]$ is used to compute the features, one of the main claims of this paper (i.e., outputs with the same prefix have similar features) is relatively trivial. To see this, suppose $x$ is an output for the system prompt $\pi$, which has length $m < n$. Then we can write $z_{\pi, x} = \frac{1}{n}\sum\_{i = 1}^{m}Z\_{\pi, x}^{\ell\_{\mathrm{last}}}[i] +  \frac{1}{n}\sum\_{i = m + 1}^{n}Z\_{\pi, x}^{\ell\_{\mathrm{last}}}[i]$; the first term is independent of $x$ and we can label it by $z_{\pi}$, while the second term can be called $z\_{\pi, x}^{r}$ (for "remainder"). Then if we compute the "centroid" $c\_{\pi}$ (again slightly ambiguous vocabulary used in the paper, but here one can take it to mean the arithmetic mean) of all outputs $x^{\prime} \in O\_{\pi}$ corresponding to system prompt $\pi$, one can write
$$c_{\pi} = \frac{1}{|O\_{\pi}|}\sum\_{x^{\prime} \in O\_{\pi}}z\_{\pi, x^{\prime}} = z_{\pi} + \frac{1}{|O\_{\pi}|}\sum_{x^{\prime} \in O\_{\pi}}z\_{\pi, x^{\prime}}^{r},$$
the second term of which we can call $z\_{\pi}^{r}$. Then for any output $x \in O\_{\pi}$ we have
$$\langle z\_{\pi, x}, c\_{\pi}\rangle = \langle z_{\pi} + z_{\pi, x}^{r}, z_{\pi} + z_{\pi}^{r}\rangle = \\|z\_{\pi}\\|^{2} + \langle z\_{\pi}, z\_{\pi}^{r} + z\_{\pi, x}^{r}\rangle + \langle z\_{\pi, x}^{r}, z\_{\pi}^{r}\rangle.$$

Now in most cases the residual input $z\_{\pi, x}^{r}$ for instance $x$ would be correlated with the average residual input $z\_{\pi}^{r}$ (unless we expect something strange to happen, e.g., the average is dominated by a few terms with all other terms pointing in the other direction, which a priori seems unlikely but I could be wrong), meaning that the last term is also positive. The middle term has unclear sign but, again, a priori the feature of the system prompt may be positively correlated (or null correlated) with the feature of the corresponding residual, so it may be positive. Thus this result follows from simple linear algebra plus a few reasonable assumptions (which would be good to check, indeed). Note that the paper mentions some normalizations before computing the cosine similarity; these normalizations manifest as multiplicative pre-factors in the above calculation and do not affect the material argument.

**Requested Changes:**

Making the paper more precise (including but not limited to the ways listed above) would be a requirement for publication. As-is, the paper is hard to evaluate for completeness and correctness.

---

### Review · Reviewer_GDY4 · 2026-02-08

**Summary Of Contributions:**

The paper shows that similar output behavior in language models does not imply the same internal control state.
By analyzing final-layer representations under different prefixes, it finds that: 1. competing or slightly modified prefixes can shift the interface-level representation state; 2. this shift may occur without any visible change in generated outputs; therefore, behavior-only evaluation is insufficient to infer whether the same control condition is maintained.

**Audience:**

Yes

**Audience Explanation:**

This paper challenges a core assumption in most of the current LLM evaluations. This is an important topic, although the key takeaway is unsurprising. Similar outputs don't guarantee similar internal representations, that's why interpretability tools are needed. On the other hand, researchers verify mechanistic interpretability methods (on the representation level) from the behavioral outputs but not in the other direction.

**Broader Impact Concerns:**

NA.

**Claims And Evidence:**

No

**Claims Explanation:**

1. The paper doesn't provide any details for the task and prompt transcript, making it extremely hard to understand the experiment setup.
2. The paper doesn't elaborate on the implications for LLM evaluations very well with evidence.

**Requested Changes:**

1. Put the Method section before Results section, otherwise the paper is hard to read.
2. Add examples, dataset details and figures to show your experiment pipelines. For example, what exact prefix you are using and what competing directions you are giving. Show the scale of your experiments: how many tasks, total prompts you have tested.
3. This is inference only task so focus more on experiments >= 7B. Update Figure 1 and Table 4 accordingly.
4. Figure 3 caption is confusing. Your color scale is labeled as cosine similarity but your caption says accuracy based on cosine similarity so I do not know what these numbers mean. Also, explain all functions in Figure 3 with examples.
5. Figure 4 why injection-only = compete but task = prefix + injection? What is competing with each other?
6. The "interface level control states" is not defined. It sounds like some behavioral output, but I don't know your task.
7. The paper would be much stronger to pick some existing common behavioral benchmark, and see how the resulting metric would change if we ran the benchmark based on the representational information.

---

### Review · Reviewer_gVRn · 2026-02-22

**Summary Of Contributions:**

This paper primarily analyze the cases where different prompts lead to the same output. The authors use different models for the simulation, but many details are missing, and the analysis raises some concerns.

**Audience:**

No

**Audience Explanation:**

This paper appears to be an unfinished work. The paper is titled The Boundary of Communication in Large Language Models, but the concept of the "Boundary of Communication" is not evident in the content. It is unclear what the authors' main findings are.

**Claims And Evidence:**

No

**Claims Explanation:**

1. The authors do not provide enough details about the simulation, such as the selection of prefixes. It is unclear which prompts and data they used.

2. The authors provide insufficient explanation of the method, for example, how the numerical results in Figures 2-4 were computed.

3. The experimental design is inadequately explained. For instance, why Figure 3 only includes results for certain models and prefixes?

4. Some definitions and explanations may be problematic. For instance, the authors don't specify what "interference" means, and what they use is not the "manifold" as most people typically understand it.

5. Some conclusions may be too strong and lack sufficient support, for example, "Fixed prefixes lead to stable final layer representations".

**Requested Changes:**

1. Please refer to the questions and concerns above.

2. The structure of the paper also needs to be rewritten, with a clearer organization that highlights the findings.

---

### Decision · Action_Editor_JfjG · 2026-04-06

**Recommendation:** Reject

**Audience:**

Yes

**Audience Explanation:**

The studied problem is relatively interesting.

**Claims And Evidence:**

No

**Claims Explanation:**

The paper investigates the identifiability of prefix control in Large Language Models (LLMs). The core premise is to examine whether behavioral similarity (similar generated outputs) implies similar underlying control states at the representation level. The authors extract final-layer representations (specifically the last token vector prior to decoding) across different prefixes and use a nearest-centroid criterion based on cosine similarity to show that prefix identity can be recovered with high accuracy.

The reviewers have raised several critical concerns:
* The draft lacked crucial methodological details, including datasets, prompt selection, and exact feature extraction pipelines.
* The use of non-standard terminology (e.g., "interface level control state," "region") made the paper difficult to follow and obscured the scientific claims.
* Reviewers pointed out that the findings, specifically that different text prefixes yield separable representations even if outputs are similar, are mathematically and intuitively expected. Reviewer pUEv provided a linear decomposition illustrating why this separation naturally occurs.
* Reviewers questioned the practical utility of the findings, asking for concrete implications or integration with standard behavioral benchmarks.

While the authors provided revisions during the rebuttal period, the reviewers reached a unanimous consensus rejection recommendation that the paper does not meet TMLR's criteria for acceptance, because the claims do not constitute a non-trivial contribution supported by sufficient empirical implications.